# Structural rather than catalytic role for mitochondrial respiratory chain supercomplexes

**Michele Brischigliaro[1,2]\*§, Alfredo Cabrera-Orefice[3], Susanne Arnold[3,4]¶,
Carlo Viscomi[1,2], Massimo Zeviani[5]#, Erika Fernández-Vizarra[1,2]\*\*\***

[1]Department of Biomedical Sciences, University of Padova, Padova, Italy; [2]Veneto Institute of Molecular Medicine, Padua, Italy; [3]Radboud Institute for Molecular Life Sciences, Radboud University Medical Center, Nijmegen, Netherlands; [4]Cologne Excellence Cluster on Cellular Stress Responses in Aging-Associated Diseases (CECAD), University of Cologne, Cologne, Germany; [5]Department of Neurosciences, University of Padova, Padova, Italy

**\*For correspondence:**
michele.brischigliaro@unipd.it
(MB);
erika.fernandezvizarra@unipd.
it (EF-V)

**Present address:** §Department
of Neurology, University of
Miami Miller School of Medicine,
Miami, United States; #Institute
for Maternal and Child Health,
IRCCS "Burlo Garofolo", Trieste,
Italy

¶Senior author

\*\*Lead contact

**Competing interest:** The authors
declare that no competing
interests exist.

**Reviewing Editor:** David Drew,
Stockholm University, Sweden

**Abstract** Mammalian mitochondrial respiratory chain (MRC) complexes are able to associate into quaternary structures named supercomplexes (SCs), which normally coexist with non-bound individual complexes. The functional significance of SCs has not been fully clarified and the debate has been centered on whether or not they confer catalytic advantages compared with the non-bound individual complexes. Mitochondrial respiratory chain organization does not seem to be conserved in all organisms. In fact, and differently from mammalian species, mitochondria from *Drosophila melanogaster* tissues are characterized by low amounts of SCs, despite the high metabolic demands and MRC activity shown by these mitochondria. Here, we show that attenuating the biogenesis of individual respiratory chain complexes was accompanied by increased formation of stable SCs, which are missing in *Drosophila melanogaster* in physiological conditions. This phenomenon was not accompanied by an increase in mitochondrial respiratory activity. Therefore, we conclude that SC formation is necessary to stabilize the complexes in suboptimal biogenesis conditions, but not for the enhancement of respiratory chain catalysis.

## eLife assessment

This study presents **valuable** findings on the organization of respiratory chain complexes in mitochondria. It provides **solid** evidence that respiratory supercomplex formation in the fruit fly does not impact respiratory function, suggesting the role of these complexes is structural, rather than catalytic. However, whether the conclusions extend to other species requires further evidence. This manuscript will be of broad interest to the field of mitochondrial bioenergetics.

## Introduction

Mitochondria are the organelles providing most of the cellular energy in form of adenosine triphosphate (ATP) in aerobic eukaryotes. The molecular machinery responsible for energy transformation is the oxidative phosphorylation (OXPHOS) system, which is canonically composed of five multiprotein complexes embedded in the inner mitochondrial membrane. OXPHOS consists of two tightly regulated processes: electron transport and ATP synthesis. Electron transport takes place between complexes I-IV and two mobile electron carriers (coenzyme Q and cytochrome *c*). During electron transport, complexes I, III, and IV pump protons from the mitochondrial matrix to the intermembrane

space, generating a proton gradient that provides the protonmotive force exploited by complex V to synthesize ATP. In mammalian mitochondria, mitochondrial respiratory chain (MRC) complexes I, III, and IV can interact with each other forming supramolecular structures described generally by the term 'supercomplexes' (*Schägger and Pfeiffer, 2000*; *Schägger and Pfeiffer, 2001*). MRC supercomplexes (SCs) can have different stoichiometries and compositions, ranging from the binding of only two complexes, such as the $I_1III_2$ and $III_2IV_1$ SCs (*Letts et al., 2019*; *Vercellino and Sazanov, 2021*), to higher order associations between complexes I, III and IV, with the SC of $I_1III_2IV_1$ stoichiometry known as the 'respirasome' (*Schägger and Pfeiffer, 2000*; *Letts et al., 2016*; *Gu et al., 2016*; *Sousa et al., 2016*; *Wu et al., 2016*; *Guo et al., 2017*). Now that the association of individual MRC complexes into supramolecular structures is well-established, with structures of several SC species being resolved, the debate is centered on what the functional significance of these structures might be.

Several possible roles have been proposed for SCs. First, it was suggested that the association between complex I (CI) and the obligate dimer of complex III ($CIII_2$) would allow substrate channeling, sequestering a dedicated coenzyme Q (CoQ) pool and allowing a more efficient electron transfer between the two complexes, while separating this electronic route from those of $FADH_2$-linked dehydrogenases (e.g. complex II) to the $CIII_2$ not bound to CI (*Schägger and Pfeiffer, 2000*; *Bianchi et al., 2004*; *Lenaz and Genova, 2007*; *Lenaz and Genova, 2009*; *Lapuente-Brun et al., 2013*; *Calvo et al., 2020*; *García-Poyatos et al., 2020*). This increased efficiency would in turn decrease electron leak and, as a consequence, produce less reactive oxygen species (ROS) than the individual free complexes (*Maranzana et al., 2013*; *Lopez-Fabuel et al., 2016*). However, the available high-resolution respirasome structures show that the distance between the CoQ binding sites in CI and $CIII_2$ are far apart and exposed to the membrane, thus not supporting the notion of substrate channeling within the SC structure (*Vercellino and Sazanov, 2021*; *Hirst, 2018*). In addition, exogenously added CoQ was necessary to sustain CI activity in the purified $I_1III_2$ SC (*Letts et al., 2019*), arguing against a tightly bound and segregated CoQ pool as a functional component of the SC. A large body of work from the late 1960s to the 1980s, resulted in the 'random collision model' to explain electron transfer between the respiratory chain complexes, and in the evidence that CoQ is present as an undifferentiated pool (*Green and Tzagoloff, 1966*; *Kröger and Klingenberg, 1973a*; *Kröger and Klingenberg, 1973b*; *Hackenbrock et al., 1986*; *Chazotte and Hackenbrock, 1988*). More recently, additional proof of the non-compartmentalized electronic routes from CI to $CIII_2$ and from complex II (CII) to $CIII_2$, came from kinetic measurements in sub-mitochondrial particles. In these systems, MRC organization in SCs was conserved but *b*- and *c*-type cytochromes in $CIII_2$ were equally accessible to CI-linked and CII-linked substrates (*Blaza et al., 2014*), and CoQ reduced by CI in the respirasomes was able to reach and readily reduce external enzymes to the SCs (*Fedor and Hirst, 2018*). In addition, growing evidence supports the notion that different MRC organizations exist in vivo, where varying proportions of SC vs. free complexes do not result in separate and distinct CI-linked and CII-linked respiratory activities (*Mourier et al., 2014*; *Lobo-Jarne et al., 2018*; *Bundgaard et al., 2020*; *Fernández-Vizarra et al., 2022*). This is in contrast with the idea that segregation into different types of SCs and in individual complexes is necessary for the functional interplay of the MRC, leading to the adaptation of the respiratory activity to different metabolic settings (*Lapuente-Brun et al., 2013*). The physical proximity of $CIII_2$ and CIV has also been suggested to promote faster electron transfer kinetics via cytochrome *c* (*Vercellino and Sazanov, 2021*; *Berndtsson et al., 2020*; *Stuchebrukhov et al., 2020*), although this is a matter of debate as well (*Trouillard et al., 2011*; *Nesci and Lenaz, 2021*).

The second main explanation to justify the existence of SCs is that they play a structural function, stabilizing the individual complexes (*Acín-Pérez et al., 2004*; *Diaz et al., 2006*) and/or serving as a platform for the efficient assembly of the complexes, with a special relevance for the biogenesis of mammalian CI (*Moreno-Lastres et al., 2012*; *Protasoni et al., 2020*; *Fernández-Vizarra and Ugalde, 2022*).

Notably, the MRC structural organization, especially the stoichiometry, arrangements and stability of the SCs, may not be conserved in all eukaryotic species (*Maldonado et al., 2021*; *Zhou et al., 2022*; *Maldonado et al., 2023*; *Klusch et al., 2023*). This is the case even within mammals, as human cells and tissues barely contain free CI, which is rather contained in SC $I_1+III_2$ and the respirasome (*Fernández-Vizarra et al., 2022*; *Protasoni et al., 2020*). In contrast, other mammalian mitochondria (bovine, ovine, rat or mouse) contain larger amounts of CI in its free form, even if the majority is still in the form of SCs (*Schägger and Pfeiffer, 2001*; *Lopez-Fabuel et al., 2016*; *Bundgaard et al.,*

*2020*; *Acín-Pérez et al., 2008*; *Letts and Sazanov, 2017*; *Davies et al., 2018*). The distribution of the MRC complexes between free complexes and SCs seems to differ even more in non-mammalian animal species. Several reptile species contain a very stable SC I+III$_2$ that lacks CIV (*Bundgaard et al., 2020*), and in *Drosophila melanogaster* practically all of CI is free, with SCs being almost completely absent (*Garcia et al., 2017*; *Shimada et al., 2018*). However, comparative studies of MRC function in diverse animal species suggest that higher amounts and stability of the SCs do not correlate with increased respiratory activity/efficiency and/or reduced ROS production (*Bundgaard et al., 2020*; *Shimada et al., 2018*).

Here we show that SCs can be stably formed in *D. melanogaster* mitochondria upon mild perturbations of individual CIV, CIII$_2$ and CI biogenesis. This finding enabled us to test whether increased SC formation translated into enhanced respiration proficiency. However, MRC performance of fruit fly mitochondria did not change regardless of the presence or absence of SCs. These observations have led us to conclude that: (1) the efficiency in the assembly of the individual complexes is likely to be the main determinant of SC formation and (2) these supramolecular complexes play a more relevant role in maintaining the stability and/or supporting the biogenesis of the MRC than in promoting catalysis.

## Results

### *D. melanogaster* MRC organization does rely on SC formation under physiological conditions

To obtain a detailed characterization of MRC organization in *D. melanogaster,* we isolated mitochondria from wild-type adults and, after solubilization with digitonin, we performed Blue-Native Gel Electrophoresis (BNGE) followed by mass spectrometry analysis of the gel lanes, using 'Complexome Profiling (CP)' (*Cabrera-Orefice et al., 2021*) and thus obtaining a profile of peptide intensities from most OXPHOS subunits along the electrophoresis lane (*Figure 1A*). This allowed us to unequivocally determine the identity of the main bands that can be visualized by Coomassie staining of BNGE gels (*Figure 1B*). The identity of the bands corresponding to complex I (CI) and complex IV (CIV) was also confirmed using specific in-gel activity (IGA) assays (*Figure 1B*). These analyses verified that *D. melanogaster* mitochondria contain extremely low amounts of high-molecular weight CI-containing SCs (*Garcia et al., 2017*; *Shimada et al., 2018*), using the same solubilization and electrophoresis conditions in which the SCs are readily detectable in mammalian cells and tissues (*Fernández-Vizarra et al., 2022*; *Acín-Pérez et al., 2008*; *Wittig et al., 2006*). As previously described (*Shimada et al., 2018*), dimeric complex V (CV$_2$) is easily visualized by BNGE and present in similar amounts as monomeric CV in *D. melanogaster* mitochondrial membranes solubilized with digitonin (*Figure 1A and B*). This CV$_2$ species is not a strongly bound dimer, as it disappears when the samples are solubilized using a harsher detergent such as dodecylmaltoside (DDM) (*Shimada et al., 2018*; *Figure 1C*). Conversely, CI is mainly found as a free complex in the native gels irrespective of whether the mitochondria had been solubilized with digitonin or DDM (*Figure 1A, B and C*). The other minor CI-containing band corresponding to the fraction associated with CIII$_2$, accounts for about 3% of the total amounts of CI and CIII$_2$, according to label-free mass spectrometry quantifications in the CP analysis (*Figure 1A*). CIV activity is absent in this band both in digitonin- and DDM- solubilized samples (*Figure 1B and C*), whereas it is present in the bands that correspond to individual CIV and dimeric CIV (CIV$_2$) detected both in digitonin- and DDM-treated samples, as well as in SC III$_2$IV$_1$, which is present only in the digitonin-solubilized samples. This is different from mammalian mitochondria in which SC III$_2$IV$_1$ is present also in DDM-solubilized mitochondria, probably due to the tight binding of CIII$_2$ to CIV through COX7A2L/SCAF1 (*Vercellino and Sazanov, 2021*; *Fernández-Vizarra et al., 2022*; *Perales-Clemente et al., 2010*). The latter does not have a homolog in *D. melanogaster* even though this species has three different COX7A isoforms (named COX7A, COX7AL, and COX7AL2) that exhibit a tissue-specific expression pattern, according to FlyBase (https://www.flybase.org/). CP analysis of *Drosophila* mitochondria only detected COX7A (mammalian COX7A1 homolog) and COX7AL2 (mammalian COX7A2 homolog), whereas COX7AL, that is solely expressed in testis, was not found. Therefore, SC I$_1$III$_2$ can be considered the only stable SC species in physiological conditions in *D. melanogaster*, yet containing a minute fraction of the total CI and CIII$_2$.

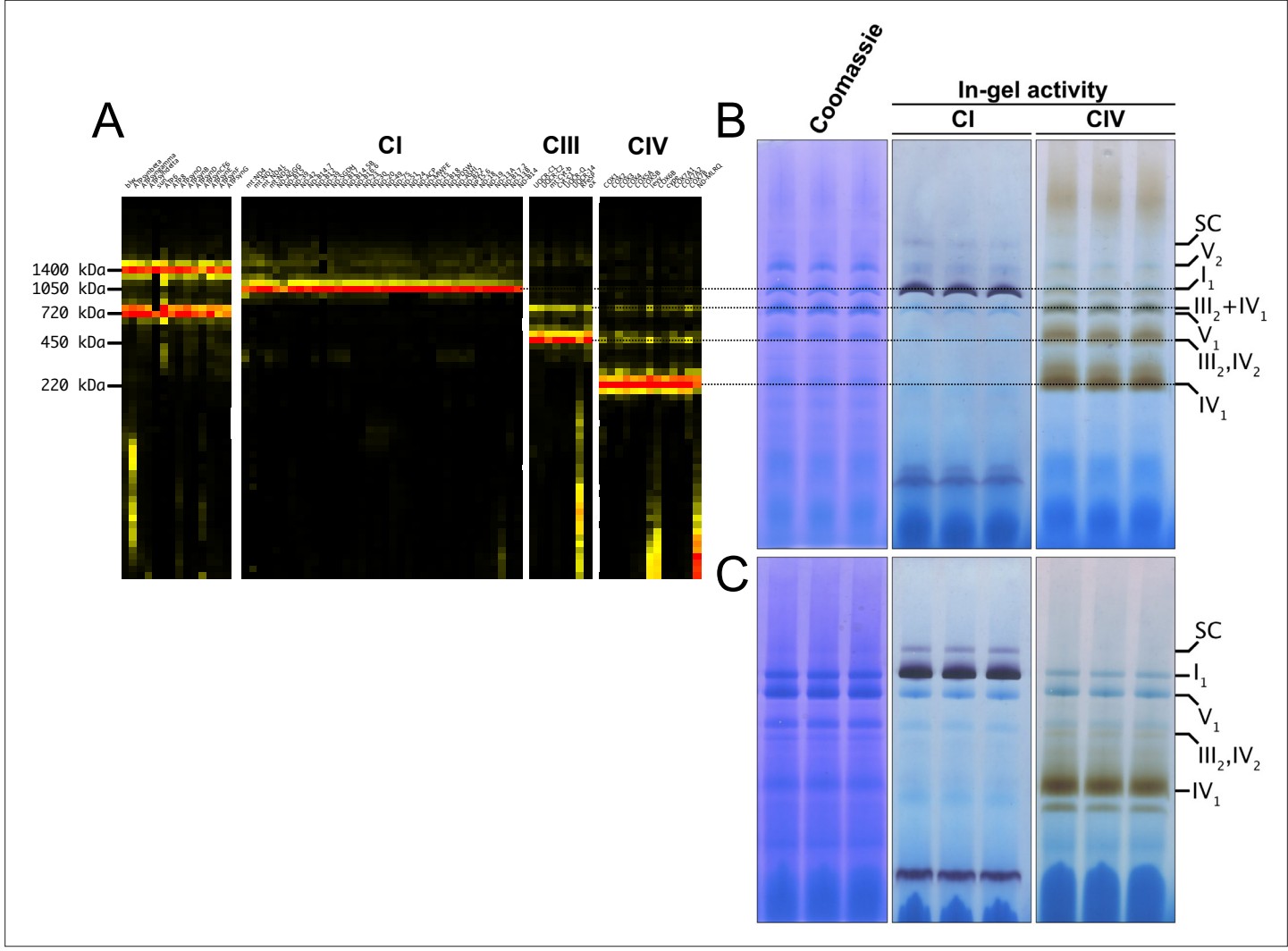

**Figure 1.** *D. melanogaster* mitochondrial respiratory chain does not rely on SC formation under physiological conditions. (**A**) Complexome profiling of wild-type *D. melanogaster* mitochondria. Heatmaps show relative abundance of MRC subunits belonging to complex I (CI), complex II (CII), complex III$_2$ (CIII), and complex IV (CIV). Color scale of normalized peptide intensities are 0 (black), 96th percentile (yellow) and 1 (red). (**B**) BN-PAGE separation of mitochondria from wild-type *D. melanogaster* solubilized with digitonin. Native gels were either stained with Coomassie R250 or analyzed by in-gel activity (IGA) for complex I (CI) and complex IV (CIV). (**C**) BN-PAGE separation of mitochondria from wild-type *D. melanogaster* solubilized with dodecylmaltoside (DDM). Native gels were either stained with Coomassie R250 or analyzed by in-gel activity assay (IGA) for complex I (CI) and complex IV (CIV).

## Perturbations of CIV assembly result in increased formation of SC I$_1$III$_2$

COA8 is a CIV assembly factor the defects of which cause isolated mitochondrial CIV deficiency in human and mouse (*Melchionda et al., 2014*; *Signes et al., 2019*), as well as in *Drosophila melanogaster* (*Brischigliaro et al., 2019*; *Brischigliaro et al., 2022a*). Consistent with the role of Coa8 in CIV biogenesis, CP analysis of mitochondria from *Coa8* knockout (*Coa8$^{KO}$*) flies showed a clear decrease in fully assembled CIV and in all the CIV-containing species (*Figure 2A and B*) when compared to the corresponding wild-type (WT) individuals (*Figure 1A* and *Figure 2B*). Curiously, CP also showed that the amounts of SC I$_1$III$_2$ were noticeably increased in the *Coa8$^{KO}$* mitochondria (*Figure 2A and B*). In these samples, complexes I and III$_2$ build a stable SC species containing ~16% of the total amount of CI, as visualized by CI-IGA, as well as by western blot (WB) and immunodetection of specific CI and CIII$_2$ subunits after BNGE in DDM-solubilized mitochondria (*Figure 2C and D*, *Figure 2—figure supplement 1*). We initially speculated that the ~fivefold increase of SC I$_1$III$_2$ formation could be

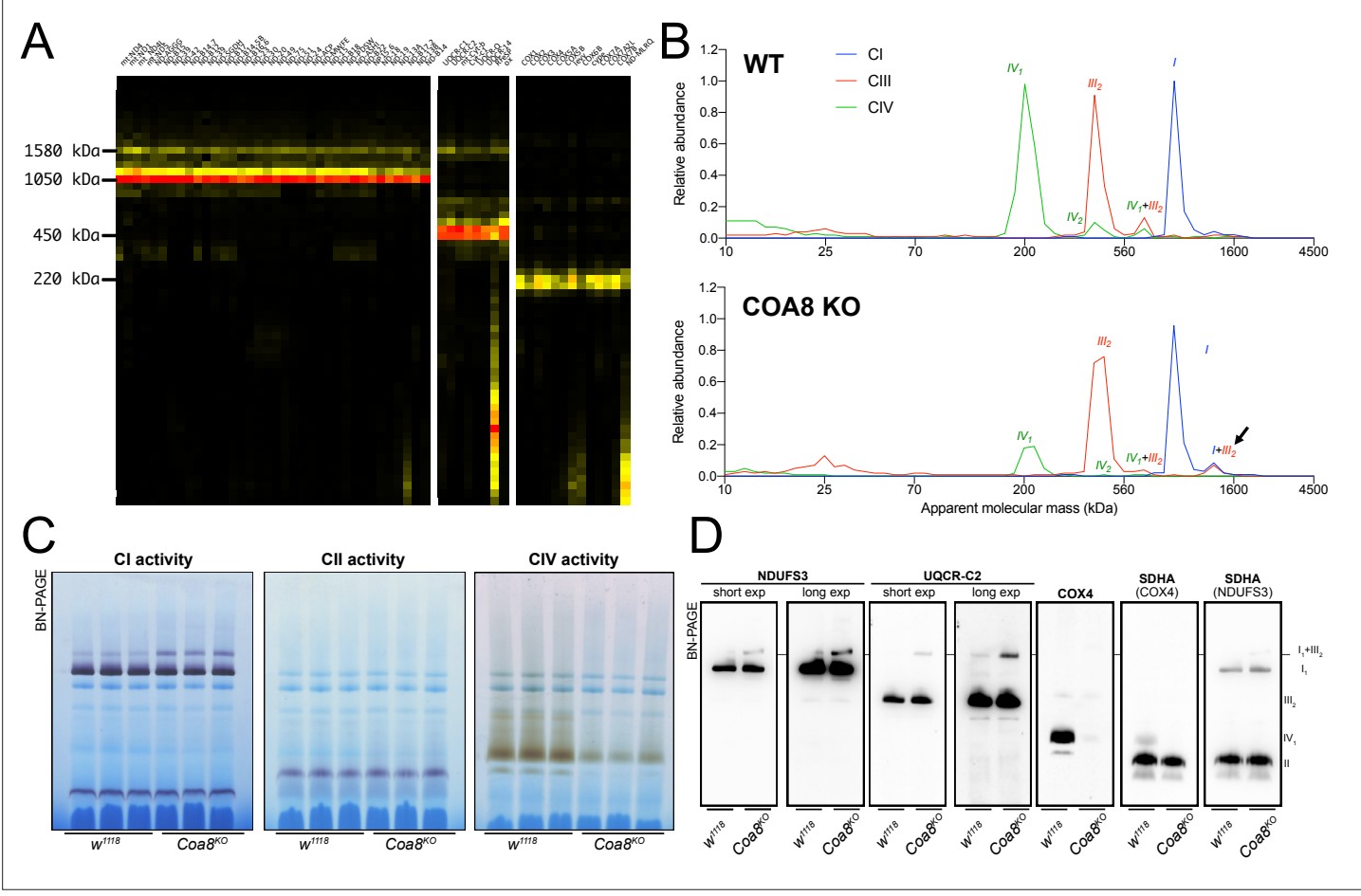

**Figure 2.** Severely perturbed CIV assembly results in increased formation of SC $I_1III_2$. (**A**) Complexome profiling of Coa8 KO *D. melanogaster* mitochondria. Heatmaps show relative abundance of MRC subunits belonging to complex I (CI), complex II (CII), complex $III_2$ (CIII) and complex IV (CIV). Color scale of normalized peptide intensities are 0 (black), 96° percentile (yellow) and 1 (red). (**B**) Average MS profiles depicted as relative abundance of MRC enzymes in natively separated complexes from wild-type (top) and Coa8 KO (bottom) fly mitochondria. Profiles of complexes I, $III_2$ and V (CI, CIII and CV) are plotted as average peptide intensity of the specific subunits identified by MS for each complex vs. apparent molecular weight. The increase in the relative abundance of SC $I_1III_2$ in Coa8 KO mitochondria is indicated by a black arrow. (**C**) In gel-activity assays for MRC complex I (CI), complex II (CII), and complex IV (CIV) in DDM-solubilized mitochondria from wild-type ($w^{1118}$) and Coa8 KO ($Coa8^{KO}$) flies. (**D**) BN-PAGE, western blot immunodetection of MRC complexes from a pool of three control wild-type ($w^{1118}$) and three Coa8 KO ($Coa8^{KO}$) fly mitochondria preparations, using antibodies against specific subunits: anti-UQCRC2 (complex III), anti-NDUFS3 (complex I), anti-COX4 (complex IV), and anti-SDHA (complex II).

The online version of this article includes the following figure supplement(s) for figure 2:

**Figure supplement 1.** Quantification of complex and supercomplex bands in $Coa8^{KO}$ fly mitochondria.

---

linked to the release of $III_2$ from SC $III_2IV_1$ induced by the strong reduction in CIV amounts when Coa8 is absent.

To test this hypothesis, we modulated Coa8 expression via the UAS/GAL4 system using RNAi driven by a 'mild' ubiquitous GAL4 driver (*da-gal4*). With this system, the *Coa8* mRNA levels were reduced to ~60% of the control (***Figure 3A***). However, these flies showed comparable levels of fully assembled CIV (***Figure 3B***, ***Figure 3—figure supplement 1***). Interestingly, in this case there was also an increased formation of SC $I_1III_2$ from ~3% in the control to ~10% in the mild *Coa8^{RNAi}* (***Figure 3B and C***, ***Figure 3—figure supplement 1***). Therefore, both strong and weak perturbations of CIV assembly produce an increased formation of CI-containing SCs in *D. melanogaster*, irrespective of whether they result in CIV deficiency or not.

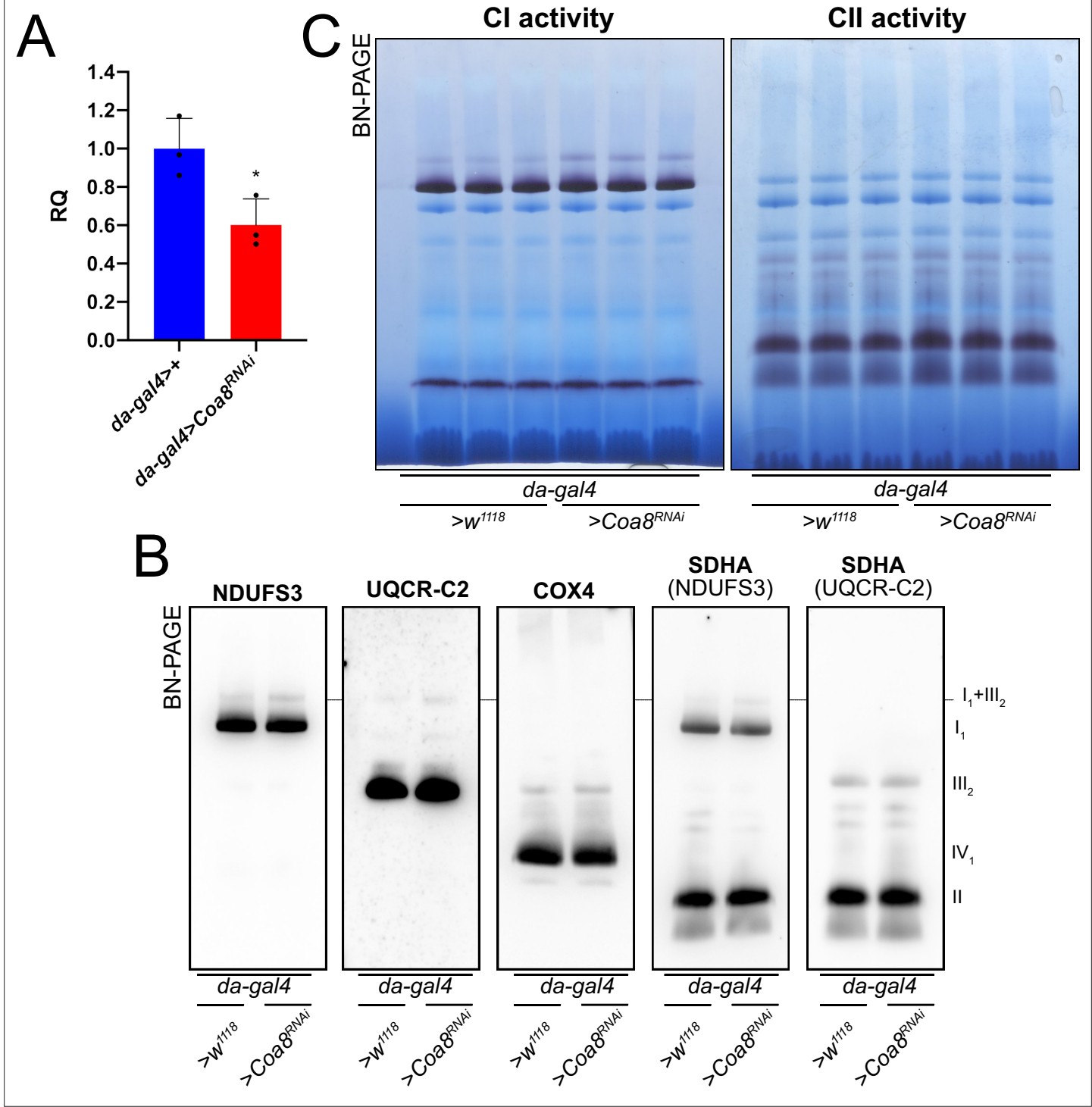

**Figure 3.** Mildly perturbed CIV assembly results in increased formation of SC $I_1III_2$. (**A**) Relative quantification (RQ) of *Coa8* mRNA expression in control (*da-gal4>+*) and *Coa8* KD (*da-gal4 >Coa8* RNAi) flies measured by qPCR. Data are plotted as mean $\pm$ SD (n = 3 biological replicates, Student's t test *p ≤ 0.05). (**B**) In gel-activity assays for MRC complex I (CI), complex II (CII) and complex IV (CIV) in DDM-solubilized mitochondria from control (*da-gal4>+*) and *Coa8* KD (*da-gal4 >Coa8* RNAi) flies. (**C**) BN-PAGE, western blot immunodetection of MRC complexes from a pool of three control (*da-gal4>+*) and three Coa8 KD (*da-gal4 >Coa8* RNAi) fly mitochondria samples, using antibodies against specific subunits: anti-UQCRC2 (complex III), anti-NDUFS3 (complex I), anti-COX4 (complex IV), and anti-SDHA (complex II).

The online version of this article includes the following figure supplement(s) for figure 3:

**Figure supplement 1.** Quantification of complex and supercomplex bands in *Coa8* KD fly mitochondria.

## Enhanced formation of SC $I_1III_2$ does not result in increased respiratory rates

SC formation was proposed to serve as a means to favor electron transfer between the complexes and therefore increase the efficiency of CI-fueled respiration (*Schägger and Pfeiffer, 2000*). With this in mind, the complete $Coa8^{KO}$ and mild $Coa8^{RNAi}$ fly mitochondria, which show increased amounts of SC $I_1III_2$ compared to the WT controls, provide an excellent opportunity to test this possibility. Oxygen consumption activities of fly homogenates in the presence of different substrates and inhibitors were analyzed by high-resolution respirometry (*Figure 4A and B*). The significant decrease in CIV enzymatic activity in the $Coa8^{KO}$ (*Figure 4C*) was not reflected by reduced oxygen consumption (*Figure 4A*). This could be explained as a result of a high CIV excess in fly mitochondria, in which the observed 60% reduction in CIV enzyme activity is still above the threshold at which the CIV defect determines lower respiratory rates (*Villani et al., 1998*; *Villani and Attardi, 2000*). In contrast, the mild reduction in *Coa8* mRNA levels did not result in CIV enzymatic deficiency but produced a slight elevation in CI activity (*Figure 4D*), which is most likely due to the increased SC $I_1III_2$ formation (*Figure 3B and C*, *Figure 3—figure supplement 1*). However, the *Coa8-KD* mitochondria did not show any differences in respiration with either CI-linked or CII-linked substrates. Also, the increased and stable interactions between complexes I and $III_2$ in the Coa8 deficient models did not produce a preferential utilization of electrons coming from CI, which would be the prediction if SC formation increased electron transfer efficiency (*Lapuente-Brun et al., 2013*).

## Mild perturbation of $CIII_2$ biogenesis also enhances SC formation in *D. melanogaster*

To determine whether increased SC $I_1III_2$ formation was specific for CIV deficient flies, we targeted $CIII_2$ by knocking down the expression of *Bcs1*. BCS1L, the human homolog, is fundamental for a correct $CIII_2$ biogenesis, being responsible for the incorporation of the catalytic subunit UQCRFS1 in the last steps of $CIII_2$ maturation (*Fernandez-Vizarra et al., 2007*; *Fernandez-Vizarra and Zeviani, 2018*). To obtain a severe $CIII_2$ defect in *D. melanogaster*, we crossed a 'strong' ubiquitous GAL4 driver (*act5c-gal4*) line with a *UAS-Bcs1* RNAi responder line (*Brischigliaro et al., 2021*). The knockdown efficiency was high, with a~75% decrease in *Bcs1* expression at the mRNA level (*Figure 5A*). In this model, *D. melanogaster* development was severely impaired causing an arrest at the larval stage (*Brischigliaro et al., 2021*). The strong $Bcs1^{RNAi}$ caused also a significant decrease in fully assembled $CIII_2$ levels (*Figure 5B and C*, *Figure 5—figure supplement 1*) and in $CIII_2$ enzymatic activity of about 50% (*Figure 5D*). Consistent with the observed $CIII_2$ deficiency, both the CI- and CII- linked respiration rates were significantly decreased by around 40% (*Figure 5E*).

In contrast, less pronounced decreases in *Bcs1* expression (*Figure 5F*) by using the mild *da-gal4* driver instead, did not produce a noticeable $CIII_2$ enzymatic defect (*Figure 5I*). However, the mild *Bcs1-KD* mitochondria showed a very different pattern of CI distribution than the controls (*Figure 5G and H*, *Figure 5—figure supplement 1*), with the formation of a 'respirasome-like' SC $I_1III_2IV_1$, and also of a higher molecular weight supercomplex (HMW-SC), of unknown stoichiometry, containing also complexes I, $III_2$ and IV, as revealed by WB and immunodetection analyses (*Figure 5G and H*). At the functional level, this was associated with a~1.5-fold increase in CI enzymatic activity (*Figure 5I*), which is proportional to the increase in total CI amounts (*Figure 5—figure supplement 1*), and in higher CI-linked respiration rates but only by ~1.2-fold (*Figure 5J*). CII-linked respiration was the same in the mild $Bcs1^{RNAi}$ samples as in the controls. Therefore, the formation of respirasome-like SCs in these mitochondria did neither increase the efficiency of electron transfer from CI nor determine a diversion of the electronic routes giving preference to the SC-bound CI.

## Mild perturbation of CI biogenesis also leads to increased SC assembly

To understand the effect of the strong and mild perturbations in CI biogenesis on MRC organization in *D. melanogaster*, we employed a similar strategy as that for CIII (see above). Crossing the strong ubiquitous *act5c-gal4* driver fly line with the *UAS-Ndufs4* RNAi responder line, produced a decrease in *Ndufs4* mRNA expression of ~90% (*Figure 6A*). Defects in *NDUFS4* are a major cause of CI deficiency-associated mitochondrial disease in humans (*Ortigoza-Escobar et al., 2016*) and the mouse and *D. melanogaster* animal models display CI deficiency and pathological phenotypes (*Kruse et al., 2008*; *Foriel et al., 2018*). Accordingly, the strong reduction in Ndufs4 expression observed in our models

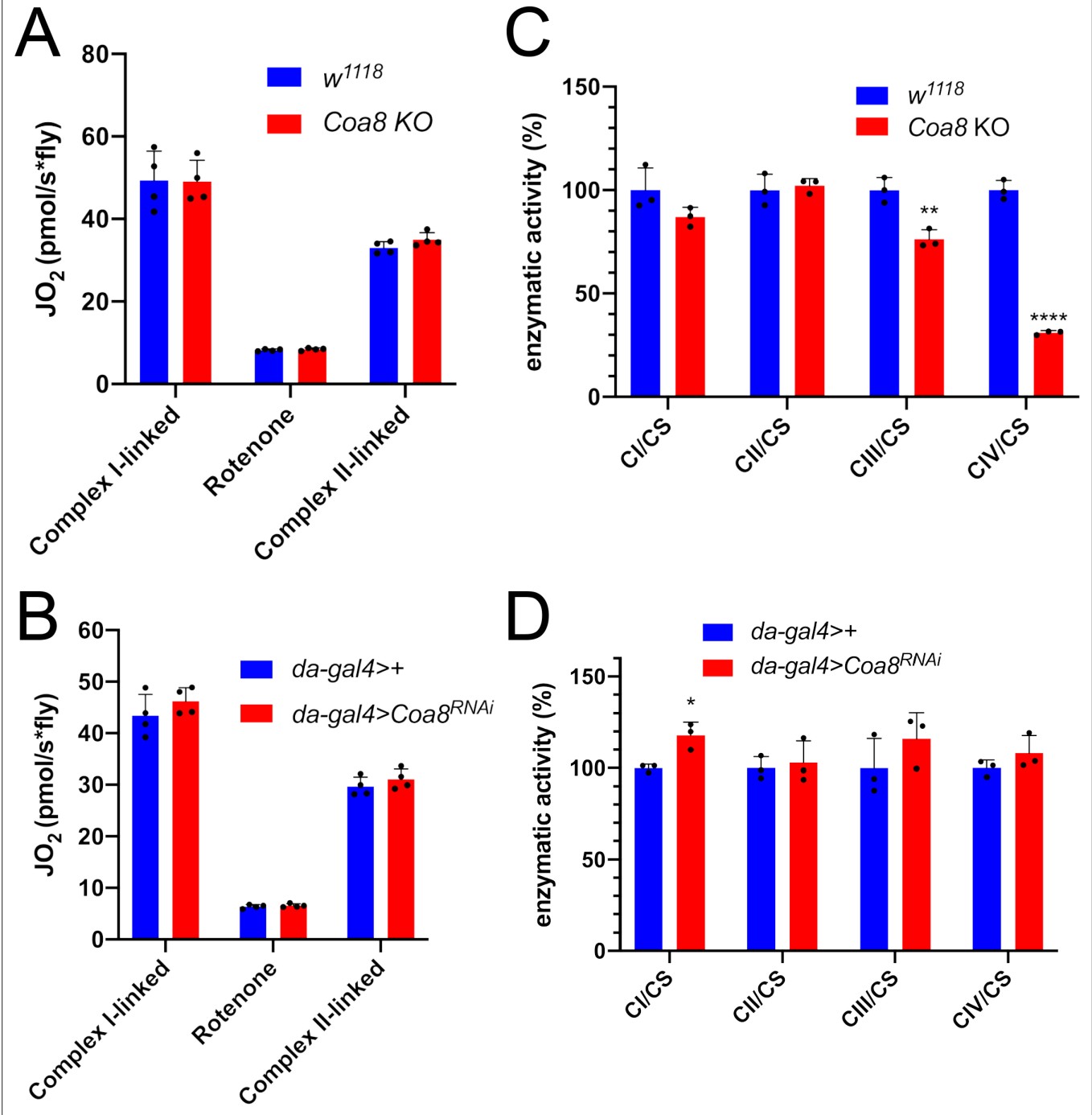

**Figure 4.** Enhanced formation of SC I$_1$III$_2$ does not result in increased respiration. (**A–B**) High-resolution respirometry (HRR) analyses of whole-fly homogenates. Respiration is represented by oxygen flux (JO$_2$) measured by oxygen consumption rates (OCR – pmol/s*fly). OCR have been measured via substrate-driven respiration under saturating concentrations of substrates inducing either complex I (CI) or complex II (CII) -linked respiration. Rotenone was used to block CI-linked respiration before measuring CII-linked respiration. HRR was performed on (**A**) *Coa8* KO and (**B**) *Coa8* KD fly homogenates compared to relative controls. Data are plotted as mean ±SD (n=4 biological replicates). (**C–D**) Kinetic enzyme activity of individual MRC complexes in (**C**) *Coa8* KO and (**D**) *Coa8* KD compared with the relative control individuals, normalized by citrate synthase (CS) activity. Data are plotted as mean ±SD (n=3 biological replicates, pairwise comparisons by unpaired Student's t test *p≤0.05, **p≤0.01, ****p≤0.0001).

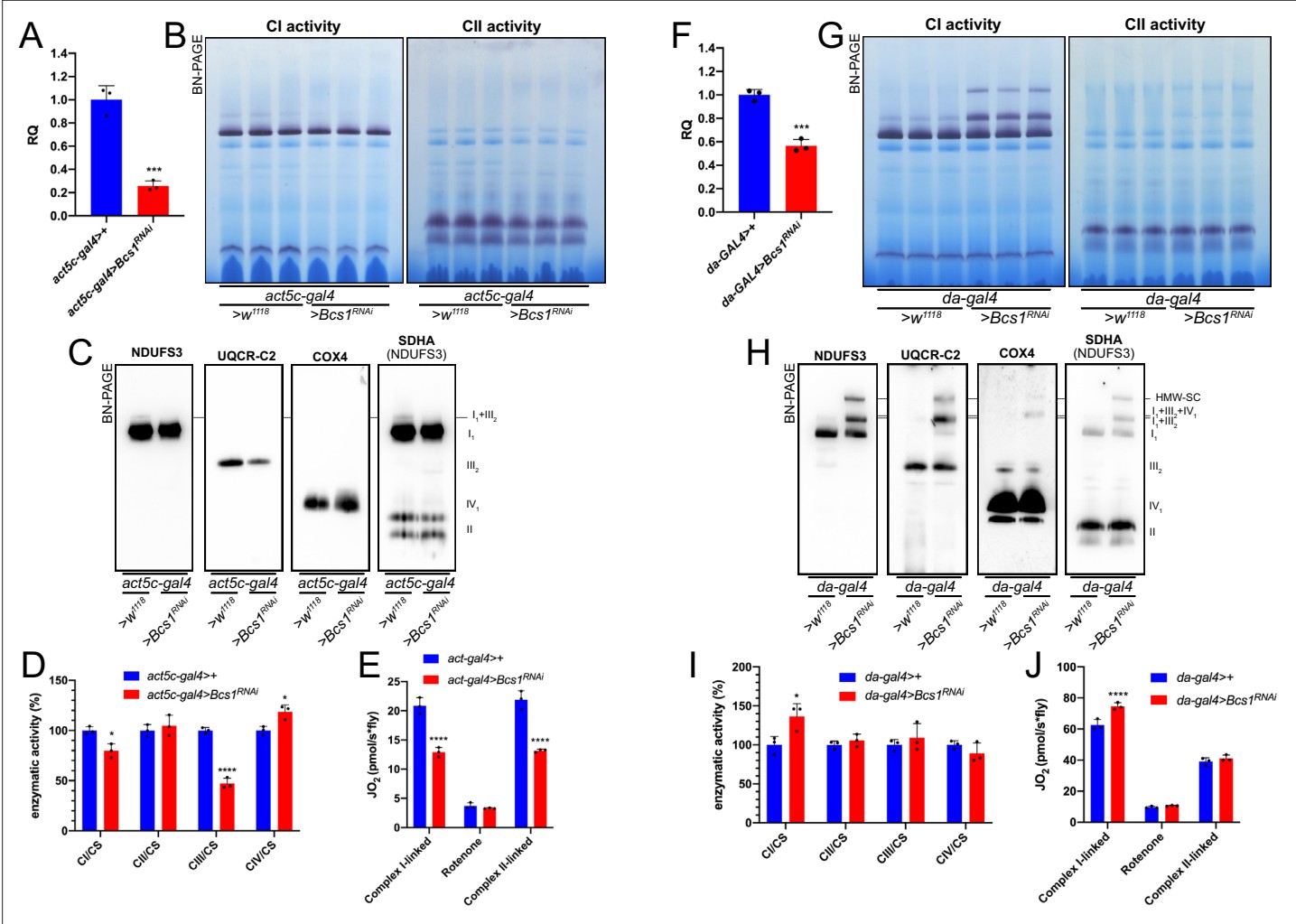

**Figure 5.** Mild perturbation of CIII₂ biogenesis enhances SC formation in *D. melanogaster*. (**A**) Relative quantification (RQ) of *Bcs1* mRNA expression in control (*act5c-gal4>+*) and *Bcs1* KD (*act5c-gal4>Bcs1* RNAi) larvae measured by qPCR. Data are plotted as mean ±SD (n = 3 biological replicates, Student's t test ***p ≤ 0.001). (**B**) In gel-activity assays for MRC complex I (CI), complex II (CII) and complex IV (CIV) in DDM-solubilized mitochondria from control (*act5c-gal4>+*) and *Bcs1* KD (*act5c-gal4>Bcs1* RNAi) larvae. (**C**) BN-PAGE, western blot immunodetection of MRC complexes from a pool of three control (*act5c-gal4>+*) and three *Bcs1* KD (*act5c-gal4>Bcs1* RNAi) larvae mitochondria samples, using antibodies against specific subunits: anti-UQCRC2 (complex III), anti-NDUFS3 (complex I), anti-COX4 (complex IV) and anti-SDHA (complex II). (**D**) Kinetic enzyme activity of individual MRC complexes in control (*act5c-gal4>+*) and *Bcs1* KD (*act5c-gal4>Bcs1* RNAi) larvae mitochondria normalized by citrate synthase (CS) activity. Data are plotted as mean ±SD (n = 3 biological replicates, pairwise comparisons by unpaired Student's t test, *p ≤ 0.05, ****p ≤ 0.0001). (**E**) High-resolution respirometry (HRR) analyses of whole-fly homogenates. Respiration is represented by oxygen flux (JO₂) measured by oxygen consumption rates (OCR – pmol/s*fly). OCR have been measured via substrate-driven respiration under saturating concentrations of substrates inducing either complex I (CI) or complex II (CII) -linked respiration. Rotenone was used to block CI-linked respiration before measuring CII-linked respiration. HRR was performed on control (*act5c-gal4>+*) and *Bcs1* KD (*act5c-gal4>Bcs1* RNAi) homogenates compared to relative controls. Data are plotted as mean ±SD (n = 3 biological replicates, two-way ANOVA with Sidak's multiple comparisons, ****p ≤ 0.0001). (**F**) Relative quantification (RQ) of *Bcs1* mRNA expression in control (*da-gal4>+*) and *Bcs1* KD (*da-gal4 >Bcs1* RNAi) larvae measured by qPCR. Data are plotted as mean ±SD (n = 3 biological replicates, Student's t test ***p ≤ 0.001). (**G**) In gel-activity assays for MRC complex I (CI), complex II (CII) and complex IV (CIV) in DDM-solubilized mitochondria from control (*da-gal4>+*) and *Bcs1* KD (*da-gal4 >Bcs1* RNAi) larvae. (**H**) BN-PAGE, western blot immunodetection of MRC complexes from a pool of three control (*da-gal4>+*) and three *Bcs1* KD (*da-gal4 >Bcs1* RNAi) larvae mitochondria samples using antibodies against specific subunits: anti-UQCRC2 (complex III), anti-NDUFS3 (complex I), anti-COX4 (complex IV), and anti-SDHA (complex II). HWM-SC: high molecular weight supercomplex. (**I**) Kinetic enzyme activity of individual MRC complexes in control (*da-gal4>+*) and *Bcs1* KD (*da-gal4 >Bcs1* RNAi) larvae mitochondria normalized by citrate synthase (CS) activity. Data are plotted as mean ±SD (n = 3 biological replicates, pairwise comparisons by unpaired Student's t test, *p ≤ 0.05). (**J**) High-resolution respirometric (HRR) analyses of whole-fly homogenates. Respiration is represented by oxygen flux (JO₂) measured by oxygen consumption rates (OCR - pmol/s*fly). OCR have been measured via substrate-driven respiration under saturating concentrations of substrates inducing either complex I (CI) or complex II (CII) -linked respiration. Rotenone was used to block CI-linked respiration before measuring CII-linked respiration. HRR was performed on

*Figure 5 continued on next page*

*Figure 5 continued*

control (*act5c-gal4>+*) and *Bcs1* KD (*act5c-gal4>Bcs1* RNAi) homogenates compared to relative controls. Data are plotted as mean ± SD (n = 3 biological replicates, two-way ANOVA with Sidak's multiple comparisons, ****p ≤ 0.0001).

The online version of this article includes the following figure supplement(s) for figure 5:

**Figure supplement 1.** Quantification of complex and supercomplex bands in *Bcs1* KD fly mitochondria.

resulted in developmental arrest and a significant decrease in fully assembled CI levels by ~40% (*Figure 6B and C*, *Figure 6—figure supplement 1*) and in a proportional decrease in NADH:CoQ oxidoreductase enzyme activity in the larvae (*Figure 6D*). BNGE analysis of the strong *Ndufs4^{RNAi} D. melanogaster* mitochondria, revealed the presence of a CI subassembly, containing the core Ndufs3 subunit (*Figure 6C*, *Figure 6—figure supplement 1*) but lacking NADH-dehydrogenase activity (*Figure 6B*). This is similar to what is observed in NDUFS4-deficient human and mouse, which accumulate the so-called ~830 kDa intermediate lacking the N-module and stabilized by the NDUFAF2 assembly factor (*Vogel et al., 2007*; *Assouline et al., 2012*; *Calvaruso et al., 2012*). In contrast, CIV levels and enzyme activity were significantly increased by 1.5-fold in the strong *Ndufs4-KD* (*Figure 6C and D*, *Figure 6—figure supplement 1*). CI-linked respiration measured in isolated mitochondria from these flies was significantly lower than in the controls, whereas the CII-linked respiration was comparable to the control (*Figure 6E*). These observations are compatible with the isolated CI defect displayed by the strong *Ndufs4-KD* flies.

Conversely, when *Ndufs4* mRNA expression was reduced to about half of the control levels (*Figure 6F*), a milder defect in CI abundance (*Figure 6—figure supplement 1*), which did not affect either NADH:CoQ oxidoreductase enzymatic activity or respiratory capacity was observed (*Figure 6G–J*) in the adult flies. However, smaller amounts of the inactive sub-CI were still detectable (*Figure 6H*). Interestingly enough, this milder perturbation of CI biogenesis by reducing the amounts of Ndufs4 also produced an increase in the formation of SC $I_1III_2$ (*Figure 6G and H*), without any changes in respiratory performance compared to the controls (*Figure 6J*).

## Discussion

The first description of MRC SCs in the early 2000s led to opposite opinions on whether these were real and functionally relevant entities. On the one hand, researchers argued that the random collision model and diffusion of the individual MRC complexes was well-established experimentally and the idea of the SCs did not fit with these observations. On the other hand, others considered that SCs were real entities and therefore they must have a functional relevance, mainly as a means to enhance electron transfer between the individual complexes. Presently, the existence of the SCs is not debated anymore, especially after the determination of the high resolution structures by cryo-electron microscopy (EM), first of the mammalian respirasomes (reviewed in *Caruana and Stroud, 2020*), followed by that of other mammalian SCs (*Letts et al., 2019*; *Vercellino and Sazanov, 2021*), and of mitochondrial respiratory chain SCs from other eukaryotic species (*Maldonado et al., 2021*; *Zhou et al., 2022*; *Maldonado et al., 2023*; *Klusch et al., 2023*; *Rathore et al., 2019*; *Hartley et al., 2019*). However, whether SCs provide any catalytical advantage to the MRC or not, is still being debated and opposing views continue to exist (*Hirst, 2018*; *Milenkovic et al., 2017*; *Hernansanz-Agustín and Enríquez, 2021*; *Cogliati et al., 2021*; *Vercellino and Sazanov, 2022*). The recent resolution of MRC SCs from different eukaryotic species has revealed that the relative arrangement of the complexes within the SCs and the bridging subunits varies substantially depending on the species. Therefore, given the conservation of the structures of the individual complexes, one could argue that if SC formation was of capital importance for MRC function, the way the complexes interact should be strictly conserved as well. Importantly, neither in the mammalian respirasomes, nor in the mammalian and plant SC $I_1III_2$ there is any evidence of substrate channeling, as the CoQ binding sites in CI and CIII$_2$ are far apart and exposed to the milieu, in principle allowing the free exchange of CoQ (*Letts et al., 2019*; *Vercellino and Sazanov, 2021*; *Hirst, 2018*; *Maldonado et al., 2023*). This is in agreement with different sets of functional data indicating that CoQ is interchangeable between the CI-containing SCs and the rest of the MRC (*Blaza et al., 2014*; *Fedor and Hirst, 2018*; *Fernández-Vizarra et al., 2022*; *Protasoni et al., 2020*). Therefore, this contrasts with the possible segmentation of the CoQ pool - one dedicated to the respirasome and the other to the FADH$_2$-linked enzymes – which has been proposed to

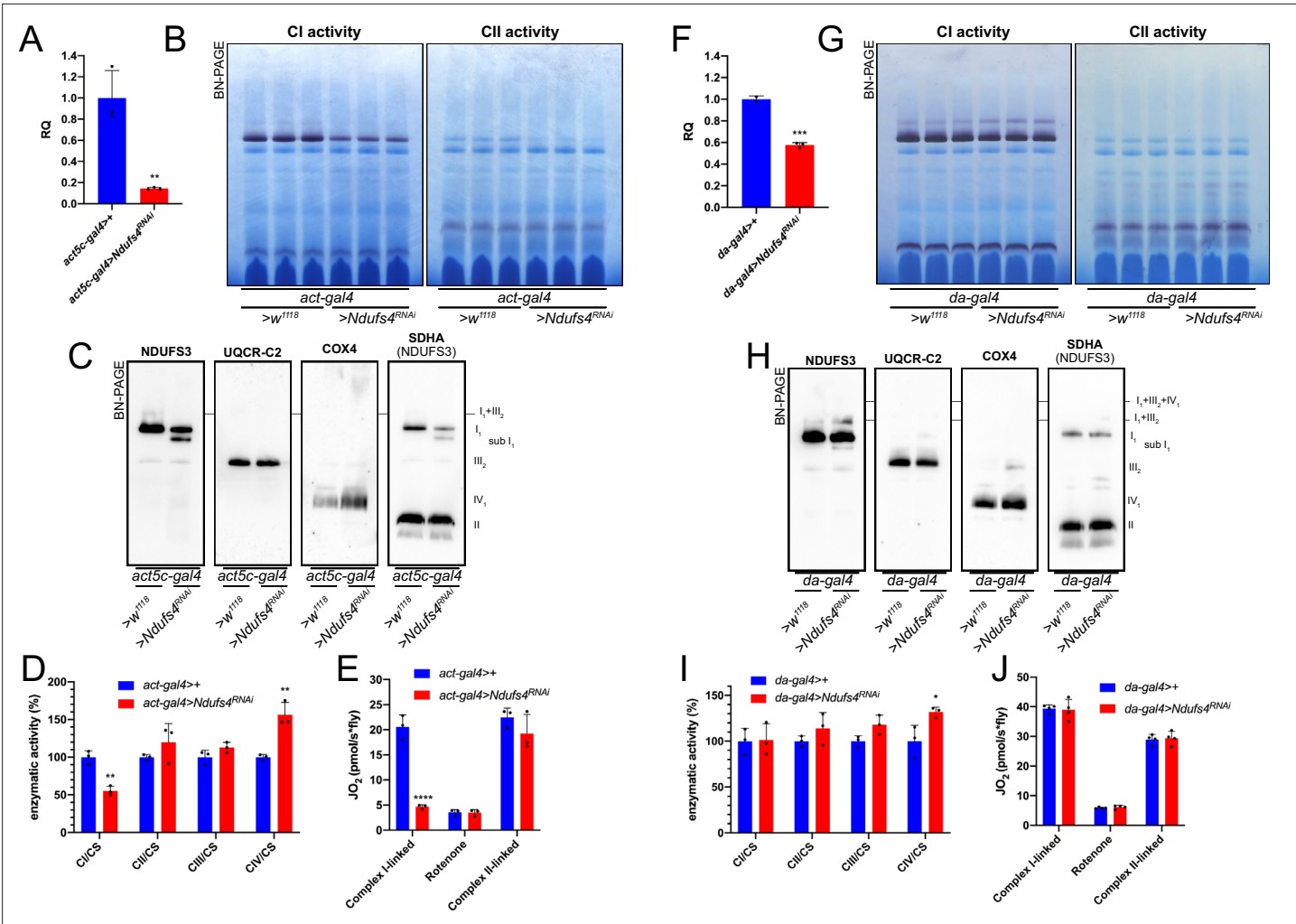

**Figure 6.** Mild perturbation of CI biogenesis enhances SC formation in *D. melanogaster*. (**A**) Relative quantification (RQ) of *Ndufs4* mRNA expression in control (*act5c-gal4>+*) and *Ndufs4* KD (*act5c-gal4>Ndufs4* RNAi) larvae measured by qPCR. Data are plotted as mean ±SD (n = 3 biological replicates, Student's t test **p ≤ 0.01). (**B**) In gel-activity assays for MRC complex I (CI), complex II (CII) and complex IV (CIV) in DDM-solubilized mitochondria from control (*act5c-gal4>+*) and *Ndufs4* KD (*act5c-gal4>Ndufs4* RNAi) larvae. (**C**) BN-PAGE, western blot immunodetection of MRC complexes from a pool of three control (*act5c-gal4>+*) and three *Ndufs4* KD (*act5c-gal4>Ndufs4* RNAi) larvae mitochondria samples, using antibodies against specific subunits: anti-UQCRC2 (complex III), anti-NDUFS3 (complex I), anti-COX4 (complex IV) and anti-SDHA (complex II). (**D**) Kinetic enzyme activity of individual MRC complexes in control (*act5c-gal4>+*) and *Ndufs4* KD (*act5c-gal4>*RNAi) larvae mitochondria normalized by citrate synthase (CS) activity. Data are plotted as mean ±SD (n = 3 biological replicates, pairwise comparisons by unpaired Student's t test, **p ≤ 0.01). (**E**) High-resolution respirometry (HRR) analyses of whole-fly homogenates. Respiration is represented by oxygen flux (JO₂) measured by oxygen consumption rates (OCR – pmol/s*fly). OCR have been measured via substrate-driven respiration under saturating concentrations of substrates inducing either complex I (CI) or complex II (CII) -linked respiration. Rotenone was used to block CI-linked respiration before measuring CII-linked respiration. HRR was performed on control (*act5c-gal4>+*) and *Ndufs4* KD (*act5c-gal4>Ndufs4* RNAi) homogenates compared to relative controls. Data are plotted as mean ±SD (n = 3 biological replicates, two-way ANOVA with Sidak's multiple comparisons, ****p ≤ 0.0001). (**F**) Relative quantification (RQ) of *Ndufs4* mRNA expression in control (*da-gal4>+*) and *Ndufs4* KD (*da-gal4 >Ndufs4* RNAi) larvae measured by qPCR. Data are plotted as mean ±SD (n = 3 biological replicates, Student's t test ***p ≤ 0.001). (**G**) In gel-activity assays for MRC complex I (CI), complex II (CII) and complex IV (CIV) in DDM-solubilized mitochondria from control (*da-gal4>+*) and *Ndufs4* KD (*da-gal4 >Ndufs4* RNAi) larvae. (**H**) BN-PAGE, western blot immunodetection of MRC complexes from a pool of three control (*da-gal4>+*) and three *Ndufs4* KD (*da-gal4 >Ndufs4* RNAi) larvae mitochondria samples, using antibodies against specific subunits: anti-UQCRC2 (complex III), anti-NDUFS3 (complex I), anti-COX4 (complex IV) and anti-SDHA (complex II). (**I**) Kinetic enzyme activity of individual MRC complexes in control (*da-gal4>+*) and *Ndufs4* KD (*da-gal4 >Ndufs4* RNAi) larvae mitochondria normalized by citrate synthase (CS) activity. Data are plotted as mean ±SD (n = 3 biological replicates, pairwise comparisons by unpaired Student's t test, *p ≤ 0.05). (**J**) High-resolution respirometry (HRR) analyses of whole-fly homogenates. Respiration is represented by oxygen flux (JO₂) measured by oxygen consumption rates (OCR - pmol/s*fly). OCR have been measured via substrate-driven respiration under saturating concentrations of substrates inducing either complex I (CI) or complex II (CII) -linked respiration. Rotenone was used to block CI-linked respiration before measuring CII-linked respiration. HRR was performed on control (*act5c-gal4>+*) and *Ndufs4* KD (*act5c-*

*Figure 6 continued on next page*

*Figure 6 continued*

gal4>*RNAi*) homogenates compared to relative controls. Data are plotted as mean ±SD (n = 4 biological replicates, two-way ANOVA with Sidak's multiple comparisons).

The online version of this article includes the following figure supplement(s) for figure 6:

**Figure supplement 1.** Quantification of complex and supercomplex bands in *Ndufs4* KD fly mitochondria.

take place as a consequence of respirasome formation (*Lenaz and Genova, 2009*; *Lapuente-Brun et al., 2013*; *Calvo et al., 2020*). A further element against the notion that SC formation is essential for mitochondrial function, is the fact that in normal conditions the MRC of *D. melanogaster* is predominantly organized based on individual complexes, as shown by us in this work and by others (*Garcia et al., 2017*; *Shimada et al., 2018*). The *Drosophila* organization was justified by a tighter packing within the mitochondrial cristae and a higher concentration of the mobile electron transporters, as a way to compensate for the lack of SCs (*Shimada et al., 2018*). However, inter-species differences in MRC organization do not appear to be of great relevance to determine the level of functionality (*Bundgaard et al., 2020*). For example, disaggregation of CI from the SCs in *A. thaliana*, where there are normally present, did not affect plant viability (*Röhricht et al., 2023*). Altogether, these observations argue against the strict requirement of SC formation to maintain MRC function.

The second main proposed role for SC is that of stabilizing and/or favoring the assembly of the individual complexes, especially that of CI (*Protasoni et al., 2020*; *Fernández-Vizarra and Ugalde, 2022*; *Lobo-Jarne and Ugalde, 2018*). Very recently, the *D. melanogaster* CI structure has been solved (*Agip et al., 2023*; *Padavannil et al., 2023*). This has provided a structural explanation as to why this complex is majorly found as an individual entity, which seems to be due the fact that the N-terminal domain of the NDUFB4 subunit, present in mammals and responsible for the interaction with CIII$_2$ within the SCs, is absent in the in the *D. melanogaster* subunit (*Padavannil et al., 2023*). Additionally, fruit fly subunit NDUFA11 has an extended C-terminus that results in a tighter binding of the subunit to the rest of the CI membrane arm. NDUFA11 stabilizes the transmembrane helix anchoring the lateral helix of subunit MT-ND5, bridging the two parts of the membrane arm of CI in plants and mammals, SC formation seem to be important for stabilizing the interaction of NDUFA11 in the CI membrane domain instead (*Letts et al., 2019*; *Letts et al., 2016*; *Maldonado et al., 2023*; *Padavannil et al., 2023*; *Padavannil et al., 2021*). However, in this work we show that there are ways to induce the formation of CI-containing SCs in fruit fly mitochondria where they normally do not exist. For example, whereas a strong impairment of CIII$_2$ biogenesis resulted in CIII$_2$ deficiency, decreased respiration and equal amounts of assembled CI, a mild perturbation of CIII$_2$ biogenesis did not result in any MRC deficiency but rather in increased CI total amounts, mainly due to enhanced SC formation. Similarly, strong decreases in Ndufs4 expression resulted in low CI amounts and activity, but milder decreases did not affect CI function although CI was redistributed into supramolecular species. In contrast, both the complete KO and mild KD of *Coa8*, induced a significantly increased formation of CI-containing SCs. Even though the *Coa8*$^{KO}$ flies display a significant reduction in CIV amounts and activity, the loss of COA8 is associated with milder phenotypes in humans, mouse and flies compared with the lack or dysfunction of other CIV assembly factors (*Melchionda et al., 2014*; *Signes et al., 2019*; *Brischigliaro et al., 2022a*). Therefore, we propose that a partial loss of CIV assembly also induces the formation of SCs in *D. melanogaster*. These observations are in line with the idea that in situations of suboptimal MRC complex biogenesis, formation of CI-containing SCs could be a way to structurally stabilize the system and preserve its function (*Calvaruso et al., 2012*; *Tropeano et al., 2020*). In the context of the cooperative assembly model, where partially assembled complexes get together before completion forming SC precursors (*Fernández-Vizarra and Ugalde, 2022*), one could envision that slower assembly kinetics would increase the chance of interactions at early stages, letting SC assembly occur in a stable way. Difference in the kinetics of CIV assembly have been observed between human and mouse fibroblasts, being slower in the human cells (*Kovářová et al., 2016*). Therefore, differences in assembly kinetics could also explain the observed different amounts and stoichiometries of the MRC SCs in different organisms. This is an interesting open question that will deserve further future investigation. In any case, this enhanced SC formation did not translate in an increase in respiratory function nor in a change in substrate preference in any of the tested models. If CI-containing SC formation enhanced respiratory activity significantly, we should have had detected a noticeable increase in CI-linked respiration in all the models of mild perturbation

of CIV, CIII$_2$ and CI. Although we did observe higher respiration with CI-linked substrates in the mild *Bcs1*-KD, this was even lower than the increase in CI enzymatic activity and total abundance.

Therefore, we conclude that the main role of SC formation is to provide structural stability to the MRC, principally for CI, rather than to enhance electron transfer between the complexes during respiration.

## Materials and methods

### Fly stocks and maintenance

Fly stocks were raised on standard cornmeal medium and kept at 23 °C, 70% humidity on a 12:12 hours light/dark cycle. Strains used in this study were obtained from Bloomington *Drosophila* Stock Center (BDSC) and Vienna *Drosophila* Resource Center (VDRC). Genotypes used in this study were: *act5c-gal4* (BDSC 4414), *da-gal4* (BDSC 8641), *UAS-Ndufs4 RNAi* (VDRC 101489), *UAS-Bcs1 RNAi* (BDSC 51863), *UAS-Coa8 RNAi* (VDRC 100605). Control strains were obtained in each experiment by crossing the specific *gal4* driver line with the genetic background flies *w$^{1118}$*. *Coa8* KO flies were generated by Wellgenetics Inc by using CRISPR/Cas9 technology, generating a 676 bp deletion, from the −49th nucleotide relative to ATG to the −34th nucleotide relative to the stop codon of Coa8.

### RNA isolation, reverse transcription, and qRT-PCR

Total RNA was extracted from 10 individuals per genotype using TRIzol (Thermo Fisher Scientific) according to the manufacturer's protocol. Reverse transcription was performed with the GoScript Reverse Transcriptase kit (Promega). qRT-PCRs were performed using GoTaq qPCR SYBR Green chemistry (Promega) and a Bio-Rad CFX 96 Touch System (Bio-Rad). The $2^{-\Delta\Delta Ct}$ method was used to calculate the expression levels of the targets (*Bcs1*, *Ndufs4*, *Coa8*) using *Rp49* as endogenous control. The oligonucleotides used are listed in the Key Resource Table.

### Isolation of mitochondria

Mitochondria from *D. melanogaster* larvae and adults were prepared by homogenization and differential centrifugation as described in *Brischigliaro et al., 2022c*. Protein concentration of mitochondrial extracts was measured with the Bio-Rad protein assay, based on the Bradford method.

### Blue-native polyacrylamide gel electrophoresis (BN-PAGE) and in-gel activity (IGA) assays

Isolated mitochondria were solubilized in 1.5 M aminocaproic acid, 50 mM Bis-Tris/HCl pH 7.0. The samples were solubilized with 4 mg of digitonin (Calbiochem) or 4 mg n-dodecyl β-D-maltoside (Sigma) per mg of protein. After 5 min of incubation on ice, samples were centrifuged at 18,000 X *g* at 4 °C for 30 min. The supernatant was collected and resuspended with Sample Buffer (750 mM aminocaproic acid, 50 mM Bis-Tris/HCl pH 7.0, 0.5 mM EDTA and 5% Serva Blue G). Native samples were separated using NativePAGE 3–12% Bis-Tris gels (Thermo Fisher Scientific) according to the manufacturer's protocol. For Coomassie staining, gels were stained with Coomassie R 250 for 20 minutes and destained/fixed using 20% methanol, 7% acetic acid. For in-gel activity assays, gels were stained with the following solutions: complex II (succinate dehydrogenase): 5 mM Tris–HCl pH 7.4, 0.2 mM phenazine methosulfate (Sigma), 20 mM succinate, and 1 mg/ml nitrotetrazolium blue chloride; Complex IV (cytochrome *c* oxidase): 50 mM potassium phosphate pH 7.4, 1 mg/ml 3′,3′-diaminobenzidine tetrahydrochloride hydrate (Sigma), 24 units/ml catalase from bovine liver (Sigma), 1 mg/ml cytochrome *c* from equine heart (Sigma), and 75 mg/ml sucrose (*Fernandez-Vizarra and Zeviani, 2021*).

### Complexome profiling

Mitochondria from *D. melanogaster* were analyzed by complexome profiling (*Brischigliaro et al., 2022b*). Isolated mitochondria (0.2 mg) were solubilized with 6 g digitonin/g protein in 50 mM NaCl, 5 mM 6-aminohexanoic acid, 1 mM EDTA, 50 mM imidazole/HCl, pH 7.0. After centrifugation at 22,000 X *g* for 20 min at 4 °C, the supernatant was supplemented with Coomassie brilliant blue G250 and proteins were separated by 4–16% gradient BN-PAGE. Digitonin-solubilized mitochondrial proteins from bovine heart were loaded as molecular mass standards. Gel lanes were cut into 60 slices, transferred to a 96-well filter microtiter plate (Millipore), and destained in 50% (v/v) methanol,

50 mM ammonium bicarbonate. After destaining, in-gel digestion with trypsin was performed. Tryptic peptides were separated by liquid chromatography and analyzed by tandem mass spectrometry (LC-MS/MS) in a Q-Exactive 2.0 Orbitrap Mass Spectrometer (2.8 SP1) equipped with an Easy nLC1000 nano-flow ultra-high-pressure liquid chromatography system (Thermo Fisher Scientific) at the front end. Thermo Scientific Xcalibur 3.1 Software Package was used for data recording. MS RAW data files were analyzed using MaxQuant (version 1.5.0.25). The extracted spectra were matched against the *Drosophila melanogaster* Uniprot Reference Sequence database (release 2020_04). Database searches were done with 20 ppm match tolerances. Trypsin was selected as the protease with two missed cleavages allowed. Dynamic modifications included N-terminal acetylation and oxidation of methionine. Cysteine carbamidomethylation was set as a fixed modification. Keratins, and trypsin were removed from the list. The abundance of each protein was determined by label-free quantification using the composite intensity based absolute quantification (iBAQ) values determined by MaxQuant analysis and was corrected for loading and MS sensitivity variations between samples based on the total iBAQ value for all detected complex V subunits. Gel migration profiles were created for each protein and normalized to the maximum abundance. Profiles of the identified mitochondrial proteins were hierarchically clustered by distance measures based on Pearson correlation coefficient (uncentered) and the average linkage method using the NOVA software package v0.5 (*Giese et al., 2015*). The visualization and analysis of the heatmaps representing the normalized abundance in each gel slice by a three-color code gradient (black/yellow/red) were done using Microsoft Excel 2019 and Graph Pad Prism 8.4.3. The mass calibration for the BN-PAGE gels was performed as previously described (*Guerrero-Castillo et al., 2017*). Membrane proteins were calibrated using the well-known molecular masses of respiratory chain complexes and supercomplexes from bovine heart mitochondria. The soluble proteins were, however, calibrated using the following set of *Drosophila* proteins: ATPB (51 kDa), malate dehydrogenase (72 kDa, dimer), citrate synthase (100 kDa, dimer), ETFA/B (122, heterodimer), heat shock protein 60 (410 kDa, heptamer), ALDH7A1 (675 kDa, dodecamer).

## Western blot and immunodetection

BN-PAGE gels were transferred to PVDF membranes in Dunn carbonate buffer (10 mM NaHCO$_3$, 3 mM Na$_2$CO$_3$) applying a constant current of 300 mA at 4 °C for 1 hr using a Mini Trans-Blot Cell (Bio-Rad). For the immunodetection of specific protein targets, blotted PVDF membranes were blocked in 5% skimmed milk in PBS-T (0.1% Tween-20) at room temperature for 1 hr and then incubated overnight with primary antibodies diluted in 3% BSA in PBS-T overnight at 4 °C. PVDF membranes were washed three times with PBS-T for 10 min, incubated with the secondary HRP- conjugated antibody for 1 hr at room temperature and washed three times with PBS-T for 10 min. Chemiluminescent signals were recorded using an Alliance Mini HD9 (UVITEC). Antibodies used are listed in the Key Resource Table. The primary antibodies against *D. melanogaster* UQCR-C2 and SdhA were a kind gift of Dr. Edward Owusu-Ansah (Columbia University, NY).

## High-resolution respirometry

To measure oxygen consumption, the flies were homogenized on ice in respiration buffer (120 mM sucrose, 50 mM KCl, 20 mM Tris-HCl, 4 mM KH$_2$PO$_4$, 2 mM MgCl$_2$, 1 mM EGTA, 1% fatty acid-free BSA, pH 7.2). Homogenates were loaded in the chamber of an O2k-HRR (High Resolution Respirometer, Oroboros Instruments) Complex I-linked respiration was measured at saturating concentrations of malate (2 mM), glutamate (10 mM), proline (10 mM) and ADP (2.5 mM). Afterwards, complex II-linked respiration was assessed adding 10 mM succinate to the measuring chamber after inhibition of complex I with rotenone (1.25 µM).

## Analysis of MRC enzymatic activities

The activities of mitochondrial respiratory chain complexes and citrate synthase (CS) were measured using kinetic spectrophotometric assays as described (*Brischigliaro et al., 2022c*).

## Statistical analysis

Statistical analysis was performed with GraphPad Prism Software, version 8.2.1. Statistical tests and significance are described in the figure captions.

## Data availability

The complexome profiling dataset containing the list of identified protein groups has been deposited in the ComplexomE profiling DAta Resource (CEDAR) repository (*van Strien et al., 2021*) with the accession number CRX45 (https://www3.cmbi.umcn.nl/cedar/browse/experiments/CRX45).

## Acknowledgements

We are grateful to Prof. Rodolfo Costa (CNR institute of Neuroscience, Padova, Italy) for providing the Coa8 KO and Bcs1 and Ndufs4 RNAi lines, Dr. Edward Owusu-Ansah (Columbia University, NY) for sharing the antibodies against *D. melanogaster* UQCR-C2 and SdhA and to Prof. Paolo Bernardi (Dept. of Biomedical Sciences, University of Padova) for critically reading the manuscript.

This research was funded by Fondazione Telethon-Cariplo Alliance GJC21014 (to EFV), Telethon Foundation GGP19007 (to MZ) and GGP20013 (to CV), AFM-Telethon 23706 (to CV), Department of Biomedical Sciences (University of Padova) FERN_FAR22_01 (to EFV) and SID2022- VISC_BIRD2222_01 (to CV), and Associazione Luigi Comini Onlus (MitoFight2, to MZ and CV).

## Additional information

### Funding

| Funder | Grant reference number | Author |
|---|---|---|
| Fondazione Telethon | GJC21014 | Erika Fernández-Vizarra |
| Fondazione Telethon | GGP19007 | Massimo Zeviani |
| Fondazione Telethon | GGP20013 | Carlo Viscomi |
| French Muscular Dystrophy Association | 23706 | Carlo Viscomi |
| Università degli Studi di Padova | FERN_FAR22_01 | Erika Fernández-Vizarra |
| Università degli Studi di Padova | VISC_BIRD2222_01 | Carlo Viscomi |

The funders had no role in study design, data collection and interpretation, or the decision to submit the work for publication.

### Author contributions

Michele Brischigliaro, Conceptualization, Data curation, Formal analysis, Investigation, Visualization, Methodology, Writing – original draft, Writing – review and editing; Alfredo Cabrera-Orefice, Data curation, Formal analysis, Investigation, Methodology, Writing – review and editing; Susanne Arnold, Formal analysis, Supervision, Methodology, Writing – review and editing; Carlo Viscomi, Supervision, Funding acquisition, Project administration; Massimo Zeviani, Conceptualization, Supervision, Funding acquisition, Project administration, Writing – review and editing; Erika Fernández-Vizarra, Conceptualization, Data curation, Formal analysis, Supervision, Funding acquisition, Visualization, Methodology, Writing – original draft, Project administration, Writing – review and editing

### Author ORCIDs

Michele Brischigliaro ⓘ https://orcid.org/0000-0003-1520-1342
Alfredo Cabrera-Orefice ⓘ http://orcid.org/0000-0002-2042-3794
Erika Fernández-Vizarra ⓘ http://orcid.org/0000-0002-2469-142X

Reviewer #1 (Public Review): https://doi.org/10.7554/eLife.88084.3.sa1
Reviewer #2 (Public Review): https://doi.org/10.7554/eLife.88084.3.sa2
Author Response https://doi.org/10.7554/eLife.88084.3.sa3

## Additional files

### Supplementary files
• MDAR checklist

### Data availability
The complexome profiling dataset containing the list of identified protein groups has been deposited in the ComplexomE profiling DAta Resource (CEDAR) repository with the accession number CRX45 (https://www3.cmbi.umcn.nl/cedar/browse/experiments/CRX45).

The following dataset was generated:

| Author(s) | Year | Dataset title | Dataset URL | Database and Identifier |
|---|---|---|---|---|
| Brischigliaro M, Cabrera-Orefice A, Arnold S, Viscomi C, Zeviani M, Fernández-Vizarra E | 2023 | Complexome profiling of *D. melanogaster* mitochondria | https://www3.cmbi. umcn.nl/cedar/ browse/experiments/ CRX45 | CEDAR, CRX45 |

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

# Appendix 1

**Appendix 1—key resources table**

| Reagent type (species) or resource | Designation | Source or reference | Identifiers | Additional information |
|---|---|---|---|---|
| Antibody | Mouse monoclonal anti-NDUFS3 | Abcam | Ab14711; RRID:AB_301429 | (1:1000) |
| Antibody | Rabbit polyclonal anti-COXIV | Cell Signaling Technology | #4844; RRID:AB_2085427 | (1:1000) |
| Antibody | Rabbit polyclonal anti-UQCR-C2 | Dr. Edward Owusu-Ansah, Columbia University, NY | N/A | (1:2000) |
| Antibody | Rabbit polyclonal anti-SdhA | Dr. Edward Owusu-Ansah, Columbia University, NY | N/A | (1:2000) |
| Sequence-based reagent | Bcs1-Fw-qPCR | This paper | qPCR primers | CTGAATGTTGCGCCAGAG |
| Sequence-based reagent | Bcs1-Rv-qPCR | This paper | qPCR primers | GACGAATGCTGCGTCGAT |
| Sequence-based reagent | Coa8-Fw-qPCR: | This paper | qPCR primers | CAATAAGCGCTTCTACGAGGA |
| Sequence-based reagent | Coa8-Rv-qPCR | This paper | qPCR primers | CCAGTTCTTGTCGAGGAACG |
| Sequence-based reagent | Ndufs4-Fw-qPCR | This paper | qPCR primers | AAGATCACCGTGCCGACTG |
| Sequence-based reagent | Ndufs4-Rv-qPCR | This paper | qPCR primers | GACAATGGGTCGCCGCTG |
| Sequence-based reagent | Rp49-Fw-qPCR | This paper | qPCR primers | ATCGGTTACGGATCGAACAA |
| Sequence-based reagent | Rp49-Rv-qPCR | This paper | qPCR primers | GACAATCTCCTTGCGCTTCT |
| Genetic reagent (*D. melanogaster*) | *D. melanogaster* strain act5c-gal4 | BDSC | 4414 | |
| Genetic reagent (*D. melanogaster*) | *D. melanogaster* strain da-gal4 | BDSC | 8641 | |
| Genetic reagent (*D. melanogaster*) | *D. melanogaster* strain UAS-Ndufs4 RNAi | VDRC | 101489 | |
| Genetic reagent (*D. melanogaster*) | *D. melanogaster* strain UAS-Bcs1 RNAi | BDSC | 51863 | |
| Genetic reagent (*D. melanogaster*) | *D. melanogaster* strain UAS-Coa8 RNAi | VDRC | 100605 | |
| Genetic reagent (*D. melanogaster*) | *D. melanogaster* strain Coa8 KO | Wellgenetics Inc | N/A | |
| Commercial assay or kit | GoScript Reverse Transcriptase Kit | Promega | A5001 | |
| Commercial assay or kit | GoTaq qPCR Master Mix | Promega | A6001 | |
| Chemical compound, drug | 3,30-Diaminobenzidine tetrahydrochloride hydrate | Merck | D5637 | |
| Chemical compound, drug | 6-Aminocaproic acid | Merck | A2504 | |

*Appendix 1 Continued on next page*

*Appendix 1 Continued*

| Reagent type (species) or resource | Designation | Source or reference | Identifiers | Additional information |
|---|---|---|---|---|
| Chemical compound, drug | Acetyl coenzyme A lithium salt | Merck | A2181 | |
| Chemical compound, drug | Antimycin A | Merck | A8674 | |
| Chemical compound, drug | BSA (fatty acid free) | Merck | A6003 | |
| Chemical compound, drug | Catalase from bovine liver | Merck | C9322 | |
| Chemical compound, drug | Coenzyme Q1 | Merck | C7956 | |
| Chemical compound, drug | Cytochrome c from equine heart | Merck | C7752 | |
| Chemical compound, drug | D-Mannitol | Merck | M4125 | |
| Chemical compound, drug | DCIP (2,6-Dichloroindophenol Sodium Salt Hydrate) | Merck | D1878 | |
| Chemical compound, drug | Decylubiquinone | Merck | D7911 | |
| Chemical compound, drug | Digitonin, High Purity | Calbiochem | 300410 | |
| Chemical compound, drug | DTNB (5,5'-Dithiobis(2-nitrobenzoic acid)) | Merck | D218200 | |
| Chemical compound, drug | EDTA (Ethylenediaminetetraacetic acid disodium salt dihydrate) | Merck | E1644 | |
| Chemical compound, drug | EGTA (Ethylene glycol-bis(2-aminoethylether)-N,N,N',N'-tetraacetic acid) | Merck | E3889 | |
| Chemical compound, drug | HEPES (N-(2-Hydroxyethyl)piperazine-N'-(2-ethanesulfonic acid), 4-(2-Hydroxyethyl)piperazine-1-ethanesulfonic acid) | Merck | H3375 | |
| Chemical compound, drug | KCN (Potassium cyanide) | Merck | 31252 | |
| Chemical compound, drug | Magnesium chloride hexahydrate | Merck | M2670 | |
| Chemical compound, drug | Malonic acid | Merck | M1296 | |
| Chemical compound, drug | n-Dodecyl-beta-maltoside (DDM) | Merck | D4641 | |
| Chemical compound, drug | NADH (β-Nicotinamide adenine dinucleotide, reduced dipotassium salt) | Merck | N4505 | |
| Chemical compound, drug | NativePAGE Cathode Buffer Additive | Thermo Fisher Scientific | BN2002 | |
| Chemical compound, drug | NativePAGE Running Buffer | Thermo Fisher Scientific | BN2001 | |
| Chemical compound, drug | Nitrotetrazolium Blue chloride | Merck | N6876 | |
| Chemical compound, drug | Oligomycin from Streptomyces diastatochromogenes | Merck | O4876 | |

*Appendix 1 Continued on next page*

*Appendix 1 Continued*

| Reagent type (species) or resource | Designation | Source or reference | Identifiers | Additional information |
|---|---|---|---|---|
| Chemical compound, drug | Oxaloacetic acid | Merck | O4126 | |
| Chemical compound, drug | Phospho(enol)pyruvic acid monopotassium salt | Merck | 860077 | |
| Chemical compound, drug | Potassium borohydride | Merck | 438472 | |
| Chemical compound, drug | Potassium phosphate dibasic | Merck | P2222 | |
| Chemical compound, drug | Potassium phosphate monobasic | Merck | P5655 | |
| Chemical compound, drug | Rotenone | Merck | R8875 | |
| Chemical compound, drug | Sodium hydrosulphite | Merck | 157953 | |
| Chemical compound, drug | Sodium succinate | Merck | S2378 | |
| Chemical compound, drug | Sucrose | Merck | S7903 | |
| Chemical compound, drug | TRIS base | Merck | T1503 | |
| Chemical compound, drug | Tween-20 | Merck | P7949 | |
| Software, algorithm | CFX Manager 3.0 | Bio-Rad | 1845000 | |
| Software, algorithm | Excel 16.69.1 | Microsoft | https://www.microsoft.com/en-us/microsoft-365/excel | |
| Software, algorithm | Graphpad Prism 8 | GraphPad Software | https://www.graphpad.com/scientific-software/prism/ | |
| Software, algorithm | Fiji v2.0.0-rc-69/1.52 p | ImageJ | https://imagej.net/software/fiji/downloads | |
| Software, algorithm | GelAnalyzer v19.1 | GelAnalyzer | http://www.gelanalyzer.com | |
| Software, algorithm | MaxQuant v1.6.10.43 | *Cox and Mann, 2008* | https://www.maxquant.org/ | |

