## [Editor Report · eLife assessment]

This study presents **valuable** findings on the organization of respiratory chain complexes in mitochondria. It provides **solid** evidence that respiratory supercomplex formation in the fruit fly does not impact respiratory function, suggesting the role of these complexes is structural, rather than catalytic. However, whether the conclusions extend to other species requires further evidence. This manuscript will be of broad interest to the field of mitochondrial bioenergetics.

---

## [Referee Report · Reviewer #1 (Public Review)]

In their manuscript, Brischigliaro et al. show that the disruption of respiratory complex assembly results in *Drosophila melanogaster* results in the accumulation of respiratory supercomplexes. Further, they show that the change in the supercomplex abundance does not impact respiratory function suggesting that the main role of supercomplex formation is structural. Overall, the manuscript is well written and the results and conclusion are supported. The *D. melanogaster* system, in which the abundance of supercomplexes can be altered through the genetic disruption of the assembly of the individual complexes, will be important for the field to discover the role of the supercomplexes. This manuscript will be of broad interest to the field of mitochondrial bioenergetics. The findings are valuable and the evidence is convincing.

Strengths

The system developed in which the relative levels of SCs can be varied will be extremely useful for studying SC physiology.

The experiments are clearly described and interpreted.

Weaknesses

The previous weaknesses identified have been addressed.

---

## [Referee Report · Reviewer #2 (Public Review)]

Respiratory chain complexes assemble in higher-ordered structures termed supercomplexes or respirasomes. The functional significance of these assemblies is currently investigated, there are two main hypothesis tested, namely that supercomplexes provide kinetic advantages or structural stability. Here, the authors use the fruitfly to reveal that, while the respiratory chain in the organism normally does not form higher-order assemblies, it does so under conditions when their assembly is impaired. Because the rather moderate increase in supercomplex formation does not change oxygen consumption stimulated by CI or CII substrate, the authors conclude that supercomplex formation has more a structural than a functional role. The main strength of this work is that the technical quality of the experiments is high and that the authors induced defects in respiratory chain assembly through sets of well-controlled genetic models. The obtained data are mostly descriptive using standard approaches and are very well executed. The authors claim that their experiments allow to conclude that the role of supercomplex formation is restricted to a structural role and, hence, exclude a function directly related to electron transport efficiency. However, while the authors can show convincingly that supercomplexes form in the mutants, but not in the wild type, the main questions still remain, namely what is the structural mechanism of supercomplex formation and what is the significance of their formation. Given that the fly system does not show supercomplex formation under normal conditions, it is likely that it evolved functionally to work different than systems having supercomplexes. Because these differences are yet unknown, it remains questionable whether the fly system can be used to inform about the general significance of supercomplexes found in the other systems.

---

## [Author Response]

The following is the authors’ response to the original reviews.

**Reviewer #1 (Public Review):**
In their manuscript, Brischigliaro et al. show that the disruption of respiratory complex assembly results in *Drosophila melanogaster* results in the accumulation of respiratory supercomplexes. Further, they show that the change in the supercomplex abundance does not impact respiratory function suggesting that the main role of supercomplex formation is structural. Overall, the manuscript is well written and the results and conclusion are supported. The *D. melanogaster* system, in which the abundance of supercomplexes can be altered through the genetic disruption of the assembly of the individual complexes, will be important for the field to discover the role of the supercomplexes. This manuscript will be of broad interest to the field of mitochondrial bioenergetics. The findings are valuable and the evidence is convincing.StrengthsThe system developed in which the relative levels of SCs can be varied will be extremely useful for studying SC physiology.The experiments are clearly described and interpreted.WeaknessesThe statement in the abstract regarding low amounts of SCs in "insect tissues" needs further support or should be narrowed. I am only aware of detailed characterization of the mitochondrial SC composition from *D. melanogaster*, which is insufficient to make a broad statement about the large and diverse category of insects. This should be rewritten.

Thank you for the comment. We have amended the text accordingly.

In the introduction (line 76) and discussion (line 283), the authors reference the CoQ binding sites in CI and CIII2 being "too far apart" to allow for substrate channeling. The distance between the active sites, though significant, is insufficient to rule out substrate channeling. A stronger argument arises from the fact that the CoQ sites of both CI and CIII2 are open to the membrane and that there are no clear barriers for the free exchange of CoQ with the membrane pool.

Thank you for the comment. We have modified both sentences accordingly.

Line 195, the slight elevation in CI amounts referred to here, does not appear to be statistically significant and therefore should not be discussed a being altered relative to the control.

To address this point of criticism we have revisited the statistical analysis, originally done by 2-way ANOVA and post-hoc test. After giving it some thought, we now consider that this might not have been the correct way to analyze either the mitochondrial respiratory chain (MRC) activity data or the densitometric quantifications. We have now used unpaired two-tailed Student’s t-test to compare the pairs of either KO or KD vs CTRL. The reason is that since the measurement of each individual MRC activity is actually an independent assay, it should be considered separately. The same applies to the densitometry because the absolute values of the intensity of individual CI and that within SCs largely differ. Therefore, we think that it is more correct to compare the abundance of individual CI in the WT vs. either KO or KD pairs and the abundance of the CI in SC independently using a t-test. With these new statistical analyses, the difference in the enzyme activity of CI reported in figure 4D is now significant, which we consider reflects better our observations. Also, with these new analyses, the difference in the amounts of CI+CIII are significantly higher in the Coa8 KD (Figure S1B). Therefore, the original affirmation is correct and we have left the sentence as it was.

Figure 4H, the assignments of the observed larger bands seem incorrect. The largest band (currently assigned as SC I1+III2+IV1) represents too large of a shift for only the addition of CIV and the band currently assigned at SC I1+III2 appears to also contain CIV. The identity of these bands should be reevaluated and additional experiments are needed to definitively prove their identity. This uncertainty should be addressed experimentally or made more explicit in the text.

Thank you for the comment. Taking a closer look at the images, we have to agree with the Reviewer that the assignment was incorrect. The higher band is too large indeed and the reviewer is correct that the band that we previously assigned as CI1+CIII2 does appear to contain CIV as well. Therefore, we have changed the labeling of that to CI1+CIII2+CIV1 because the stoichiometry is compatible with the apparent MW. Also, we have renamed the higher MW band to HMW-SC (high-MW SC) of uncertain nature (unknown stoichiometry) but clearly containing all three complexes I, III and IV. We amended the text (lines 219-221) plus figures 5H and S1 accordingly.

Line 302, the authors state that the structural basis for less SC in *D. melanogaster* is "due to a more stable association of the NDUFA11 subunit..." However, this would not result is a less stable SC association and only explains why NDUFA11 is more stably associated with CI in the absence of CIII2. The more likely structural reason for the observation of less SC in *D. melanogaster* is the N-terminal truncation of Dm-NDUFB4 relative to mammalian NDUFB4. This truncation results in the loss of a major SC interaction site between CI and CIII2 in the matrix.

Thank you for pointing this out. We have amended the text accordingly.

**Reviewer #2 (Public Review):**
Respiratory chain complexes assemble in higher-ordered structures termed supercomplexes or respirasomes. The functional significance of these assemblies is currently investigated, there are two main hypothesis tested, namely that supercomplexes provide kinetic advantages or structural stability. Here, the authors use the fruitfly to reveal that, while the respiratoy chain in the organism normally does not form higher-order assemblies, it does so under conditions when their assembly is impaired. Because the rather moderate increase in supercomplex formation does not change oxygen consumption stimulated by CI or CII substrate, the authors conclude that supercomplex formation has more a structural than a functional role. The main strength of this work is that the technical quality of the experiments is high and that the authors induced defects in respiratory chain assembly through sets of well-controlled genetic models. The obtained data are mostly descriptive using standard approaches and are very well executed. The authors claim that their experiments allow to conclude that the role of supercomplex formation is restricted to a structural role and, hence, exclude a function directly related to electron transport efficiency. However, while the authors can show convincingly that supercomplexes form in the mutants, but not in the wild type, their main claim is not well supported by data and both the structural mechanism of supercompelx formation and their significance remain unknown. While the supercomplex formation observed only in mitochondrial mutants per se is interesting, it would be good to great to define structural aspects of supercomplex formation and their potential impact on the stability of the respiratory chain complexes in these mutants.

We thank the Reviewer for the positive assessment of our work and the suggestions to improve the manuscript.

**Reviewer #1 (Recommendations For The Authors):**
The sentence on line 90, which starts "This is in contrast with..." is unclear and needs to be rewritten.

Thank you. We have modified the sentence to make it clearer.

Lines 153 and 155, reference is made to tissue specific expression patterns but no literature reference is provided.

Thank you for the comment. The tissue specific expression patterns of the different isoforms are reported in the FlyBase database. We added the link to website in the text.

Line 188, "...homogenates in presence of..." should read "homogenates in the presence of..."

Thank you. Amended.

Line 336, "...lower to the increase..." should read "...lower than the increase..."

Thank you. Amended.

**Reviewer #2 (Recommendations For The Authors):**
In order to unravel the molecular mechanism by which supercomplexes form in the mutant, it would be important to identify the factor mediating this. Prime candidates would be additional proteins that co-purify of co-fractionate with the respiratory chain when they assemble into supercomplexes or changes in the lipid composition of the mitochondria, where cardiolipin has been shown to stabilize supercomplex formation. The inclusion and analysis of complexome data for all mutants would be excellent, plus an MS analysis of a purified supercomplex.

Thank you for the suggestion to which we completely agree. We have taken a closer look to the hierarchical clustering of peptide intensities in our complexome profiling data, which clusters the proteins according to their similarity in electrophoretic migration within the complexes. We have specifically looked for proteins in which the peptide intensity changed in a similar fashion as the complex I structural subunits. Among the four candidate proteins (Uniprot IDs Q8SXY6, Q95T19, Q9W0Y6, Q9VJQ3), only Q95T19 — Serine--tRNA synthetase-like protein Slimp is annotated as a mitochondrial protein. This protein is a *Drosophila*-specific paralog of the mitochondrial Serine-tRNA synthetase generated by gene duplication (PMID: 20870726), which carries out a function linking mitochondrial translation with mtDNA maintenance (PMID: 30943413). Therefore, in principle we would not consider it as a good candidate to be a ‘SC assembly factor’. The identification of factors promoting the formation of SC in *Drosophila* under these conditions is definitely an important point warranting future investigation.

The authors could define the stability of the respiratory chain complexes through metabolic pulse-chase labeling experiments. This could reveal that the role of supercomplex formation is indeed structural, improving stability.

We agree that this would be an important piece of information to understand the phenomenon we have observed. Unfortunately, it is technically impossible to perform metabolic labeling of mitochondrial proteins in whole flies. It would be possible to perform in organello pulse-chase labelling, however our previous experience indicates that complex I does not completely assemble de novo in isolated mitochondria (PMID: 20385768).

The authors should analyze oxygen consumption from mitochondria isolated from larvae as in the other experiments on enzyme activities or the (high-quality) BN-PAGE, and not from whole flies that are homogenized. Moreover, they need to determine the quantities of the complexes by complementary experiments (MS, Western blotting or spectroscopy).

Thank you for the comments. However, we believe that repeating the entire analyses with the larvae would not add significant information to the work and the main interpretation would not change, as the main claim of the paper is based on the data collected on adult flies. In addition, the band patterns of MRC complexes in the BNGE is the same between larvae and adults and therefore, does not depend on the developmental stage.Regarding the quantification of the complexes, we think that the data provided by using complementary approaches such as in gel activity assays (IGA), western blot (WB) and kinetic assays of MRC enzymatic activities, allowed us to confidently determine the amount of the individual complexes. Hence, we performed IGA assays and enzymatic activity assays (which reflect the amounts of fully assembled and functional complexes) in triplicate (independent samples). For the WB analyses, due to the scarcity of some of the antibodies available to detect the Dm MRC proteins, which were a kind gift of Dr. Edward Owusu-Ansah (Columbia University), we decided to pool the three independent samples of each group before running them through the Blue-Native gels. The densitometric curves of the WB bands (Figure S2) show the abundance of each individual MRC complex within the ‘free’ and SC forms. We prioritized the BN analyses over SDS-PAGE and WB analysis, as we consider that just measuring the steady-state levels of MRC subunits is not as informative, because it is possible that certain subunits are present in the mitochondrial membranes but not assembled into the final mature structures.

Can changes in Coenzyme Q levels explain the absence of a defect on electron transport? This could be determined for the mutant as well as the wild type animals.

We agree that this would be a relevant aspect to investigate. For example, determining whether lower CoQ levels are able to maintain the same respiratory activities in the models in which higher amounts of SCs are formed, as it was proposed in Shimada et al. (PMID: 29191512) would be very interesting. However, the fact that the mild KD models show no MRC enzymatic defects whatsoever (Figure 4D, Figure 5I and Figure 6I), provides the most straightforward explanation to the observed absence of respiratory defects.